# UFGTIME: REFORMING THE PURE GRAPH PARADIGM FOR MULTIVARIATE TIME SERIES FORECASTING IN THE FREQUENCY DOMAIN

## ABSTRACT

Recent advances in multivariate time series forecasting have seen a shift toward a pure graph paradigm, which transforms time series into hypervariate graphs and employs graph neural networks (GNNs) to holistically capture intertwined spatiotemporal dependencies. While promising, this approach faces notable challenges. First, converting time series into hypervariate graphs often neglects essential temporal sequences, which are vital for accurately capturing temporal dependencies. Second, treating the graph as a complete structure can obscure the varying importance of intra- and inter-series connections, potentially overlooking key local patterns. To address these challenges, we introduce a novel hyperspectral graph data structure that embeds sequential order into frequency signals and employs a sparse yet meaningful topological structure. In addition, we propose the UFGTIME framework, featuring a frequency-based global graph framelet message-passing operator tailored to hyperspectral graphs, effectively mitigating the smoothing issue and capturing global insights through sparse connections. Extensive experiments demonstrate that our framework significantly surpasses state-of-the-art methods, excelling in both short- and long-range time series forecasting while achieving superior efficiency. Our code is available at: `https://anonymous.4open.science/r/UFGTIME-E352`.

## 1 INTRODUCTION

Multivariate time series forecasting is crucial in industrial applications such as transportation, manufacturing, and energy management. Recent advancements in deep neural networks, evolving from recurrent to convolutional and attention-based models, have significantly improved forecasting accuracy. However, many of these methods fail to fully capture the critical spatial correlations, which are essential for modeling complex dependencies in multivariate time series data. Graph Neural Networks (GNNs) have emerged as a powerful approach to address this limitation. Initially, GNNs were employed to capture spatial information across time steps, which was then combined with historical temporal information through a forecasting model. This approach led to foundational models such as DCRNN (Li et al., 2018), GraphWaveNet (Wu et al., 2019), STGCN (Yu et al., 2018), and MTGNN (Wu et al., 2020). Despite their success, these models often rely on integrating GNN modules with separate components to capture temporal dependencies, treating the processes of modeling spatial and temporal dependencies as independent tasks. This separation contradicts the intertwined nature of spatial temporal information.

Recent research, FourierGNN (Yi et al., 2024), has explicitly tackled these intertwined interactions by transforming time series into a new data structure called the **hypervariate graph**, pioneering a novel pure graph paradigm for multivariate time series forecasting. However, this pure graph-based method faces two key challenges:

**Challenge 1.** *Processing multivariate time series in a hypervariate graph structure risks losing essential sequential information.* **Challenge 2.** *The fully connected setting in hypervariate graphs tends to reduce attention to various connections, which leads to overlooking important temporal patterns.*

To address these challenges, we propose an innovative frequency graph framelet time series forecasting framework, named UFGTIME. Our method transforms the time series into frequency domain signals, preserving the sequential information of the original time series within the frequency signals. We further reform a novel graph data structure called the ***hyperspectral graph***, which transforms frequency signals into graph features and constructs sparse topological structures based on signal similarities to enhance attention to cross-signal relationships. Finally, we propose a global manner framelet message-passing operator to capture global patterns through sparse graph connections and mitigate the smoothing effects caused by aggregation between similar nodes. Our contributions are as follows:

- Identifying the limitations of hypervariate graphs in ignoring temporal sequential information and neglecting attention to local connection patterns in fully connected graphs setting.
- Proposing an innovative hyperspectral graph structure, reorganizing signals from fast Fourier transforms, preserving temporal sequential order in frequency signals, and using KNN to capture local (sparse) connections between signal features.
- Introducing a global framelet message-passing operator to capture the global patterns of the hyperspectral graph through sparse connections and alleviate the smoothing effects from connecting similar nodes.
- Validating the effectiveness of the UFGTIME framework through extensive experiments, demonstrating its ability to outperform state-of-the-art methods.

## 2 PRELIMINARIES AND RELATED WORKS

### 2.1 PROBLEM DEFINITION

A multivariate time series $\boldsymbol{X} \in \mathbb{R}^{N \times T \times D}$ represents a sequence of $D$-dimensional vector observations of $N$ entities recorded over a time period $T$. Given a window size $T$, we denote $\boldsymbol{X}_t = [X_{t-T+1}, \ldots, X_{t-1}, X_t] \in \mathbb{R}^{N \times T \times D}$ representing the observations on the looking back-window of size $T$ at time $t$, where $X_t \in \mathbb{R}^{N \times D}$ is the observation for all the $N$ entities at $t$. A typical forecasting task is to learn a model $f(\cdot)$, by minimizing a predefined loss function, such that

$$\widehat{\boldsymbol{Y}}_{t+1} = f(\boldsymbol{X}_t) = f([X_{t-T+1}, \ldots, X_t]) \tag{1}$$

predicts the next $\tau$ steps of another time series $\boldsymbol{Y}$ at $t+1$, e.g., $\boldsymbol{Y}_{t+1} = [X_{t+1}, \ldots, X_{t+\tau}] \in \mathbb{R}^{N \times \tau \times D}$, forecasting within the prediction time window $\tau$.

### 2.2 PARADIGM OF GNNS IN MULTIVARIATE TIME SERIES FORECASTING

Deep learning methods such as convolutional neural networks (CNNs) (Borovykh et al., 2017; Assaf et al., 2019), recurrent neural networks (RNNs) (Connor et al., 1994; Hochreiter & Schmidhuber, 1997), and transformers (Zhou et al., 2021; Zhang & Yan, 2023) have demonstrated considerable success in multivariate time series forecasting. However, a significant limitation of these approaches lies in their inability to explicitly model the spatial topological structure information. To address the challenges in temporal dynamics, GNNs have been applied using different paradigms, which can be categorized as follows:

**Modularity Paradigm** The introduction of DCRNN (Li et al., 2018), which integrates graph and recurrent modules into an end-to-end framework, marked a significant advancement in capturing spatial correlations along with temporal dynamics. This design has become the dominant paradigm for applying GNNs in multivariate time series forecasting. Variants of this approach have combined GNNs with different architectures, leading to methods such as *GNNs with recurrence* (e.g., ST-MetaNet (Pan et al., 2019), STGNN (Wang et al., 2020), AGCRN (Bai et al., 2020), GTS (Shang et al., 2021), and HiGP (Cini et al., 2024)), *GNNs with convolution* (e.g., GraphWaveNet (Wu et al., 2019), MTGNN (Wu et al., 2020), StemGNN (Cao et al., 2020), STGODE (Fang et al., 2021), MTGODE (Jin et al., 2022), and CaST (Xia et al., 2024)), and *GNNs with temporal attention* (e.g., GMAN (Zheng et al., 2020), STAR (Yu et al., 2020), and TPGNN (Liu et al., 2022)). Despite their effectiveness, these frameworks often treat spatial correlation and temporal processes as separate entities, which may result in a disjointed representation of spatio-temporal dependencies, potentially misrepresenting the inherent interconnections found in real-world scenarios.

**Pure Graph Paradigm**   To address the limitations of the modularity paradigm in capturing the complex entanglement of spatial and temporal information, the FourierGNN (Yi et al., 2024) introduces the pure graph paradigm, which transforms multivariate time series into a data structure known as a ***hypervariate graph***:

**Definition 1** (**Hypervariate Graph**). *Given a general multivariate time window $\boldsymbol{X}_t \in \mathbb{R}^{N \times T \times D}$ at timestamp $t$, a hypervariate graph is defined as $\boldsymbol{G}_t^H = \left(\boldsymbol{X}_t^G, \boldsymbol{J}\right)$, where $\boldsymbol{X}_t^G \in \mathbb{R}^{NT \times D}$ represents the node features, and $\boldsymbol{J} \in \mathbf{1}^{NT \times NT}$ denotes the fully connected adjacency matrix.*

By transforming the multivariate time series into a pure graph structure, each timestamp entity in the multivariate time series is represented as a node within the hypervariate graph, with all entities fully connected. This approach inherently embeds both temporal dynamics and spatial correlations within the graph, enabling the forecasting problem to be reformulated as:

$$\widehat{\boldsymbol{Y}}_{t+1} = g(\boldsymbol{G}_t^H) = g(\boldsymbol{X}_t^G, \boldsymbol{J}), \tag{2}$$

where $g(\cdot)$ is a GNN that accepts node features and adjacency matrices as inputs. This paradigm emphasizes a fully integrated graph representation of spatial and temporal data, offering significant potential for time series analysis. However, the hypervariate graph method has certain **limitations**, such as the ***potential oversight orders of the time sequences*** and ***the challenges posed by the fully connected graph***, which will be discussed in Section 3.

## 3   LIMITATIONS OF STATES-OF-ARTS WORKS IN PURE GRAPH PARADIGM

### 3.1   OVERSIGHT OF TIME SERIES SEQUENTIAL ORDER

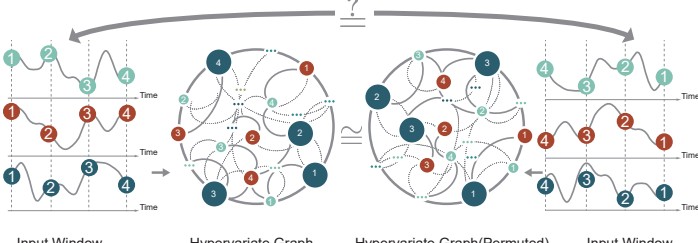

| PERM | MAE | RMSE | MAPE |
|------|------|------|------|
| None | 0.0565 | 0.0879 | 0.1366 |
| 25% | 0.0677 | 0.0988 | 0.1471 |
| 50% | 0.0749 | 0.1075 | 0.1849 |
| 75% | 0.0767 | 0.1085 | 0.1891 |
| 100% | 0.0778 | 0.1114 | 0.1936 |

Figure 1: A demonstration of Losing Sequential information on Hypervariate Graph

Table 1:   Permutations on Hypervariate Graph

Recent work on FourierGNN utilizes hypervariate graphs to integrate temporal dependencies and spatial information, proposing a novel pure graph paradigm for time series forecasting. However, this approach overlooks a critical aspect: the sequential order inherent in temporal data, which is essential for capturing time-dependent relationships. In hypervariate graphs, the topological structure is typically assumed to be fully connected, with each node representing temporal features at a specific timestamp. Permuting the node order in a hypervariate graph results in an isomorphic graph, indicating the hypervariate graph's invariance to node ordering. Figure 3.1 visualizes two multivariate time series with identical values but different sequential orders, forming isomorphic hypervariate graphs, despite the underlying time series being fundamentally different due to the order change. Our findings underscore the limitations of hypervariate graphs in preserving sequential order, as formalized in the following proposition (with proof provided in Appendix A.1):

**Propostion 1.** *The hypervariate graph is insensitive to the temporal order of the original time series, thereby discarding critical sequential information.*

We conducted an empirical study to assess the performance of FourierGNN under time series order permutations. The results, as shown in Table 1, indicate limited performance variation, even with large-scale permutations on the temporal dimension of the data. This suggests that the FourierGNN model in hypervariate graphs may be relatively insensitive to temporal sequential order, implying that hypervariate graphs might not effectively preserve sequential information.

### 3.2   FULL CONNECTION REDUCE ATTENTION OF HYPERVARIATE GRAPH

In the hypervariate graph setting, as defined in Definition 1, the graph topology is represented by a fully connected adjacency matrix, indicating that all vertices are uniformly connected. In

this context, each node in the hypervariate graph corresponds to a specific timestamp within a multivariate time series. This fully connected design aims to capture the comprehensive dynamics of global time dependence. However, many studies have demonstrated that local patterns within time series can be equally significant (Papadimitriou & Yu, 2006; Chen et al., 2006). Due to the uniform distribution of connections, a fully connected graph tends to diminish the importance of local connections in the hypervariate graph, potentially overlooking some meaningful implicit local patterns. To support our argument, we constructed a toy example of a hypervariate graph involving a multivariate temporal system with four time series, each containing nine timestamps.

We computed a sparse Laplacian matrix using the K-Nearest Neighbors (KNN) algorithm based on the temporal characteristics. We compared it to a fully connected Laplacian, as illustrated in Figure 2. The sparse Laplacian reveals significant attention patterns in intra- and inter-series connections, whereas the fully connected Laplacian uniformly distributes attention across all connections, diminishing their relative importance. Additionally, fully connected graphs have several other drawbacks, such as a tendency to over-smooth (Huang et al., 2020) and higher computational complexity, but these limitations are beyond the scope of this work.

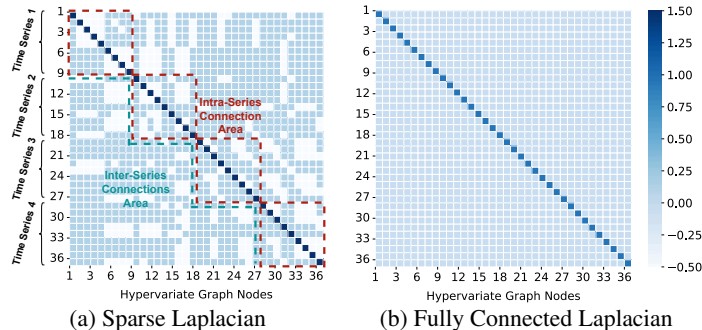

(a) Sparse Laplacian  (b) Fully Connected Laplacian

Figure 2: Visualization of Laplacian Patterns in a Toy Hypervariate Graph. (a) The Sparse Laplacian is generated using the KNN algorithm, showing distinct attention to intra- and inter-connection nodes. (b) The Fully Connected Graph demonstrates uniform connections across all nodes within the hypervariate graph, highlighting neglect attention across the network.

## 4 PROPOSED METHOD

### 4.1 BYPASS SEQUENTIAL ORDER OF TIME SERIES WITH DOMAIN TRANSFER

As discussed in Section 3.1, state-of-the-art approaches within the pure graph paradigm may suffer from overlooking the sequential order inherent in time series due to the transformation into a hypervariate graph. A promising strategy to address this limitation involves representing the multivariate time series $\boldsymbol{X} \in \mathbb{R}^{N \times T \times D}$ in the spectral domain using orthogonal bases, expressed as $\boldsymbol{S} = \mathcal{F}(\boldsymbol{X}) \in \mathbb{C}^{N \times C \times D}$, where $\mathcal{F}(\cdot)$ denotes the Fourier transform applied along the time dimension, and $C$ represents the length of the transformed temporal signal in the spectral domain. Subsequently, a graph is constructed using the spectral signals $\boldsymbol{S}$ following the method that forms the hypervariate graph. This new type of graph is defined as follows:

**Definition 2 (Hyperspectral Graph).** *Given a general multivariate time window $\boldsymbol{X}_t \in \mathbb{R}^{N \times T \times D}$ at time step $t$, the spectral temporal signals $\boldsymbol{S}_t$ are defined as the Fourier-transformed time series, $\boldsymbol{S}_t = \mathcal{F}(\boldsymbol{X}_t) \in \mathbb{C}^{N \times C \times D}$. The hyperspectral graph $\boldsymbol{G}_t^S$ at timestamp $t$ is then defined as $\boldsymbol{G}_t^S = \left(\boldsymbol{S}_t^{\boldsymbol{G}}, \boldsymbol{A}_t\right)$, where $\boldsymbol{S}_t^{\boldsymbol{G}} \in \mathbb{C}^{NC \times D}$ represents the graph features, and $\boldsymbol{A}_t \in \{0, 1\}^{NC \times NC}$ is the adjacency matrix associate with hyperspectral graph features $\boldsymbol{S}_t^{\boldsymbol{G}}$.*

Our motivation behind advocating using hyperspectral graphs lies in the inherent flexibility of the Fourier transformation in interpreting and manipulating the frequency components derived from a signal (e.g., a 1D discrete (ordered) signal in the case of Discrete Fourier transform). The Fourier transformation output can be interpreted as frequency components contained in the original signal. Although mathematically all the frequency components are inherently presented in a specific order from the lowest to the highest frequencies, this order is not part of signal information but the amount of different frequencies. Thus the Fourier coefficients can be rearranged according to the analysis's needs without impacting the data's integrity. As long as the frequency associated with each output component is clearly understood and consistently applied, the actual order of these components can be adjusted to suit specific analytical goals. This flexibility indicates that it is appropriate to use graphs to represent output data from the Fourier transform. Therefore, transforming the multivariate

time series into a hyperspectral graph can effectively alleviate the issue of overlooking the temporal order present in the hypervariate graph.

## 4.2 Propagating Features via Hyperspectral Graph

Unlike the hypervariate graph introduced in Definition 1, where fully connected graphs are constructed across all timestamps, the adjacency matrices in the hyperspectral graph are estimated based on signal similarity using algorithms KNN. Generally, this can be expressed as $A_t = \text{KNN}\left(\text{Re}(S_t^G)^\frown\text{Im}(S_t^G), k\right)$, where $k$ represents the number of neighbors, and $\text{Re}(S_t^G)^\frown\text{Im}(S_t^G)$ denotes the concatenation of the real and imaginary components of the hyperspectral graph features. Once the graph structure is defined, the next challenge is to process the features $S^G$ with the generated sparse graph structure $A_t$. Since $A_t$ is sparse, employing traditional GNNs to propagate hyperspectral graph features may limit the focus to local (short-sightedness), nearest-neighbor connections rather than capturing global relationships. Additionally, because $A_t$ is constructed from the top $k$ signal similarities in the hyperspectral graph, propagating the features risks over-smoothing. This can lead to embedding becoming indistinguishable after propagation, which is detrimental for downstream tasks such as forecasting. Therefore, a desired GNN model for hyperspectral graphs must satisfy the following requirements:

> 1. *Prevent excessive smoothing (similarity) of hyperspectral graph features by preserving the identifiability (sharpness) of each node's features during propagation.*
> 2. *Propagate node features in a global manner, even though the graph is sparsely connected.*

**How to select GNN for hyperspectral graphs?**  GNNs that satisfy *Requirement 1* often employ a diffusion-reaction paradigm, where node features are first homogenized through spatial propagation (i.e., using $A$). Then the ego-graph feature (e.g., $S_t^G$) is added to reintroduce variation into the system (Choi et al., 2023; Han et al., 2024; Thorpe et al., 2022). While these models have achieved remarkable results, spectral GNNs, such as ChebNet (Defferrard et al., 2016), typically learn filtering functions (e.g., diagonal matrices) in the spectral domain (i.e., the eigenspace of the graph Laplacian), which enables feature propagation from a global perspective, thus satisfying *Requirement 2*. Consequently, an ideal model would either be a spectral GNN that can induce multiple feature dynamics or a spatial GNN that accounts for global dependencies between features. In light of these considerations, we focus on a family of spectral GNNs known as **Graph Framelets**, which meet the aforementioned requirements (Zheng et al., 2021).

## 4.3 Graph Framelets on the Fourier domain

In this section, we formulate a novel graph framelet system, called **UFGTIME**, specifically designed for hyperspectral graphs in multivariate time series forecasting tasks, addressing the unique demand posed by this setting. While numerous framelet variants have been developed in recent years (Shi et al., 2023; Han et al., 2022; Liu et al., 2023), these approaches primarily focus on node features in the real domain. In contrast, our work reforms the original graph framelet framework (Zheng et al., 2021; Yang et al., 2022) to the spectral domain of hyperspectral graphs. This new design allows us to better capture the intricate relationships of the hyperspectral graph.

**Graph Framelet and Framelet Message-Passing**  Graph framelets are defined by a set of filter banks, denoted as $\eta_{a,b} = \{a; b^{(1)}, \ldots, b^{(L)}\}$, and the corresponding complex-valued scaling functions. These scaling functions are typically expressed as $\Psi = \{\alpha; \beta^{(1)}, \ldots, \beta^{(L)}\}$, where $L$ represents the number of high-pass filters. The framelet framework adheres to the following refinement relationship between the scaling functions and filter banks:

$$\widehat{\alpha}(2\xi) = \widehat{a}(\xi)\widehat{\alpha}(\xi) \quad \text{and} \quad \widehat{\beta^{(r)}}(2\xi) = \widehat{b^{(r)}}(\xi)\widehat{\alpha}(\xi), \ \forall \xi \in \mathbb{R}, \ r = 1, \ldots, L,$$

where $\widehat{\alpha}$ and $\widehat{\beta^{(r)}}$ denote the Fourier transforms of $\alpha$ and $\beta^{(r)}$, respectively, and $\widehat{a}, \widehat{b^{(r)}}$ represent the Fourier series of $a$ and $b^{(r)}$. The graph framelets are then defined as

$$\varphi_{j,p}(v) = \sum_{i=1}^{n} \widehat{\alpha}\left(\frac{\Lambda_i}{2^j}\right) u_i(p)u_i(v) \text{ and } \psi_{j,p}^r(v) = \sum_{i=1}^{n} \widehat{\beta^{(r)}}\left(\frac{\Lambda_i}{2^j}\right) u_i(p)u_i(v),$$

for $r = 1, \ldots, L$ and scale level $j = 1, \ldots, J$. Here, $u_i(v)$ refers to the eigenvector $\boldsymbol{u}_i$ at node $v$. The functions $\varphi_{j,p}(\cdot)$ and $\psi_{j,p}^r(\cdot)$ are commonly referred to as the *low-pass framelets* and *high-pass framelets* at node $p$. One can define the framelet decomposition matrices $\mathcal{W}_{0,J}$ and $\mathcal{W}_{r,J}$ as:

$$\mathcal{W}_{0,J} = \boldsymbol{U}\widehat{a}\left(\frac{\boldsymbol{\Lambda}}{2^{m+J}}\right)\cdots\widehat{a}\left(\frac{\boldsymbol{\Lambda}}{2^m}\right)\boldsymbol{U}^\top, \mathcal{W}_{r,0} = \boldsymbol{U}\widehat{b^{(r)}}\left(\frac{\boldsymbol{\Lambda}}{2^m}\right)\boldsymbol{U}^\top, \quad \text{for } r = 1, \ldots, L, \quad (3)$$

$$\mathcal{W}_{r,j} = \boldsymbol{U}\widehat{b^{(r)}}\left(\frac{\boldsymbol{\Lambda}}{2^{m+j}}\right)\widehat{a}\left(\frac{\boldsymbol{\Lambda}}{2^{m+j-1}}\right)\cdots\widehat{a}\left(\frac{\boldsymbol{\Lambda}}{2^m}\right)\boldsymbol{U}^\top, \quad \text{for } r = 1, \ldots, L, \, j = 1, \ldots, J. \quad (4)$$

Here, $m$ represents the coarsest scale level, which is the smallest value satisfying $2^{-m}\lambda_N \leq \pi$. It can be shown that $\sum_{(r,j)\in\mathcal{I}} \mathcal{W}_{r,j}^\top \mathcal{W}_{r,j} = \boldsymbol{I}$ for $\mathcal{I} = \{(r,j) : r = 1, \ldots, L, \, j = 0, 1, \ldots, J\} \cup \{(0, J)\}$, indicating the *tightness* of the framelet decomposition and reconstruction. We highlight that in real practice, to avoid heavy eigen-decomposition of the graph Laplacian, one may adopt the $K$ order polynomial to boost the implementation speed (refer to Appendix B.1). In summary, one can explicitly denote the (spectral) feature propagation of the graph framelet (without activation) as

$$\boldsymbol{S}_t^{\boldsymbol{G}}(\ell + 1) = \sum_{(r,j)\in\mathcal{I}} \mathcal{W}_{r,j}^\top \text{diag}(\boldsymbol{\theta}_{r,j}) \mathcal{W}_{r,j} \boldsymbol{S}_t^{\boldsymbol{G}}(\ell) \boldsymbol{W}(\ell), \quad (5)$$

where $\text{diag}(\boldsymbol{\theta}) \in \mathbb{R}^{NT \times NT}$ contains learnable coefficients in each frequency domain and $\boldsymbol{W}(\ell)$ is the weight matrix that is shared across the different frequency domains. One can check that the dynamic in Equation 5 propagates the node features in a global view due to its spectral filtering nature. In addition, we expect our framelet model to maintain the global spectral filtering manner with relatively low computational cost. We, therefore, adopt the framelet message-passing manner (Liu et al., 2023), which suggests that the reconstruction process of the graph framelet can be omitted. Resulting as

$$\boldsymbol{S}_t^{\boldsymbol{G}}(\ell + 1) = \sum_{(r,j)\in\mathcal{I}} \mathcal{W}_{r,j} \boldsymbol{S}_t^{\boldsymbol{G}}(\ell) \boldsymbol{W}_{r,j}(\ell), \quad (6)$$

suggesting a distinguished channel-mixing operation, i.e., $\boldsymbol{W}_{r,j}(\ell) \in \mathbb{C}^{D \times D}$ across different frequency domains. However, we expect our model to remain in the global spectral filtering manner but also with comparable complexity. We propose an innovative framelet message-passing by aligning the same channel-mixing operation to all frequency domains with different filtering coefficients. Accordingly, we have

$$\boldsymbol{S}_t^{\boldsymbol{G}}(\ell + 1) = \sum_{(r,j)\in\mathcal{I}} \text{diag}(\boldsymbol{\theta}_{r,j}) \mathcal{W}_{r,j} \boldsymbol{S}_t^{\boldsymbol{G}}(\ell) \boldsymbol{W}(\ell) \quad (7)$$

**Theoretical Analysis** To show graph framelet meets *Requirement 1*, let us consider the framelet model with Haar filtering function of scale one. That is, when $J = 1$, we have:

$$\mathcal{W}_{0,1} = \boldsymbol{U}\boldsymbol{\Lambda}_{0,1}\boldsymbol{U}^\top = \boldsymbol{U}\cos(\boldsymbol{\Lambda}/8)\boldsymbol{U}^\top, \quad \mathcal{W}_{1,1} = \boldsymbol{U}\boldsymbol{\Lambda}_{1,1}\boldsymbol{U}^\top = \boldsymbol{U}\sin(\boldsymbol{\Lambda}/8)\boldsymbol{U}^\top, \quad (8)$$

and the following one-layer framelet convolution from Equation 5 (without activation) can be further denoted as:

$$\boldsymbol{S}_t^{\boldsymbol{G}}(\ell + 1) = \left(\boldsymbol{U}\text{diag}(\boldsymbol{\theta}_{0,1})\cos^2(\boldsymbol{\Lambda}/8) + \text{diag}(\boldsymbol{\theta}_{1,1})\sin^2(\boldsymbol{\Lambda}/8)\boldsymbol{U}^\top\right)\boldsymbol{S}_t^{\boldsymbol{G}}(\ell)\boldsymbol{W}(\ell). \quad (9)$$

When we fix $\boldsymbol{\theta}_{0,1} = \boldsymbol{1}^{NT}$, where $\boldsymbol{1}^{NT}$ is the vector of all ones, and $\boldsymbol{\theta}_{1,1} = \theta\boldsymbol{1}^{NT}$, one can check that when $\theta > 1$, the model is dominated by the high-pass filtering dynamic, i.e., $\text{diag}(\boldsymbol{\theta}_{1,1})\sin^2(\boldsymbol{\Lambda}/8)$ since the function $\sin^2(\boldsymbol{\Lambda}/8)$ is monotonically increasing over the graph spectral domain. On the other hand, when $0 < \theta < 1$, the model dynamic is dominated by the low-pass filtering dynamic, i.e., $\text{diag}(\boldsymbol{\theta}_{0,1})\cos^2(\boldsymbol{\Lambda}/8)$ as $\cos^2(\boldsymbol{\Lambda}/8)$ is monotonically decreasing via the spectral domain. This shows graph framelets can naturally induce both smoothing and sharpening dynamics to meet the requirement . It is worth noting that our conclusion can be smoothly applied to our proposed model in equation 7. We refer the more theoretical details to the works in (Shi et al., 2023; Shao et al., 2023b).

**Complexity Analysis** For simplicity of analysis, we assume that the weights of our Fourier domain framelet message-passing operator are of the form $\mathbb{C}^{D \times D}$, and the frequency signal has the same length $T$ as the time series. The time complexity of a single layer of our framelet operator is $\mathcal{O}(NTkD + NTD^2)$, where $k$ represents the number of neighbours in the KNN graph. More details on the complexity analysis can be found in Appendix B.3.

Figure 3: A workflow demonstration of UFGTIME framework for predicting $\widehat{Y}_{t+1}$ with input $X_t$

## 4.4 MULTIVARIATE TIME SERIES FORECASTING WITH UFGTIME

The main framework of UFGTIME is illustrated in Figure 3. Given input multivariate time series data $X_t \in \mathbb{R}^{N \times T \times D}$, we first apply moving-average decomposition to the input to extract trend information $T_t \in \mathbb{R}^{N \times T \times D}$, and then apply the Fast Fourier Transform (FFT) on the time dimension of the input to obtain the frequency signal $S_t \in \mathbb{C}^{N \times C \times D}$. The frequency signal is reshaped into a hyperspectral graph feature $S_t^G \in \mathbb{C}^{NC \times D}$. Next, we leverage KNN to generate a sparse topological structure $A_t \in \{0, 1\}^{NT \times NT}$, associated with the input $\mathrm{Re}(S_t^G) ^\frown \mathrm{Im}(S_t^G)$. At this point, we obtain the hyperspectral graph $G_t^S = (S_t^G, A_t)$. Subsequently, to capture intricate dependencies on the hyperspectral graph, we feed the data into $\ell$ layers of a global framelet message-passing operator with a SiLU activation function, defined as $S_t^G(\ell + 1) = \mathrm{SiLU} \left( \sum_{(r,j) \in \mathcal{I}} \mathrm{diag}(\theta_{r,j}) \mathcal{W}_{r,j} S_t^G(\ell) W(\ell) \right)$.

Afterward, we reshape $S_t^G(\ell+1)$ into frequency signal $S_t(\ell+1) \in \mathbb{C}^{N \times C \times D}$ and use the Inverse Fast Fourier Transform (IFFT) $\mathcal{F}^{-1}(S_t(\ell+1))$ to obtain the output hidden state $H_t \in \mathbb{R}^{N \times T \times D}$. Finally, based on the output hidden state $H_t$, which encodes spatiotemporal interdependencies, we apply a two-layer feed-forward network (FFN) (see Appendix B.2) to project it onto $\tau$ future steps. This result is combined with the trend embedding to yield the final output $\widehat{Y}_{t+1} = \mathrm{FFN}(H_t) \oplus \mathrm{Lin}(T_t) \in \mathbb{R}^{N \times \tau \times D}$.

## 5 EMPIRICAL EVALUATION

### 5.1 EXPERIMENTAL SETUP

**Datasets** We employ several datasets for short-term multivariate time series forecasting, including SOLAR-FL, WIKI-500, TRAFFIC, ECG, ELECTRICITY2H, and COVID-CAL. For long-term multivariate time series forecasting, we utilize the ETTM1, ETTM2, ETTH1, and ETTH2 datasets. We adopt the original data splits provided by Yi et al. (2024) and Zhou et al. (2021) to ensure a fair comparison. Additional details and data sources are in Appendix B.4.

**Baselines** Our baselines encompass a range of well-established models in time series forecasting, which can be classified into three categories. 1) Transformer-based methods: Autoformer (Wu et al., 2021), Informer (Zhou et al., 2021), Pyraformer (Liu et al., 2021), and Crossformer (Zhang & Yan, 2023). 2) Graph-based methods: DCRNN (Li et al., 2018), STGCN (Yu et al., 2018), GWNet (Wu et al., 2019), MTGNN (Wu et al., 2020), StemGNN (Cao et al., 2020), AGCRN (Bai et al., 2020), and FourierGNN (Yi et al., 2024). 3) Linear-based methods: DLinear (Zeng et al., 2023), and TiDE (Das et al., 2023). Additional baseline details can be found in Appendix B.6.

**Implementation** We reproduce the baseline models using revised scripts from FourierGNN (Yi et al., 2024) and the fair benchmarking toolkit BasicTS+ (Shao et al., 2023a). The models are fine-tuned using Adam and RMSprop optimizers to minimize the MSE loss. Additional fine-tuning details can be found in Appendix B.6.

### 5.2 OVERALL PERFORMANCE ANALYSIS

***Can* UFGTIME *effectively capture temporal patterns from short input sequences?*** The partial performance of short-term multivariate time series forecasting is presented in Table 2 (full performance results shown in Table 6), where both the history window and forecasting length are set to 12. The best results are highlighted in grey. It is important to note that some transformer-based methods, such as Autoformer, Informer, and Pyraformer, do not produce results on the ECG

dataset due to the lack of time-stamp information. The key observations from the experiments are summarized as follows. Compared to all state-of-the-art baselines, UFGTIME demonstrates exceptional performance across short-term forecasting datasets. Notably, on the COVID-CAL, SOLAR-FL, and WIKI-500 datasets, UFGTIME shows a strong ability to capture complex dynamic patterns that often challenge transformer-based methods. This highlights the effectiveness of the dedicated multi-resolution graph framelet architecture and underscores the importance of managing sequential order in pure graph paradigms for multivariate time series forecasting. For other datasets, which exhibit clearly distinguishable seasonal patterns, all baselines, including UFGTIME, perform at a similar level, demonstrating the generalization capability of our method for typical time series datasets.

Table 2: Short-Term Multivariate Time Series Forecasting Results on Four Datasets. Best and Second Best Results Per Dataset Highlighted in Grey and Underlined Respectively. **ECG** Results for Partial Transformer-based Methods are Denoted as '−' Due to Missing Temporal Information.

| BASELINES | SOLAR-FL | | | WIKI-500 | | | ECG | | | COVID-CAL | | |
|---|---|---|---|---|---|---|---|---|---|---|---|---|
| | MAE | RMSE | MAPE | MAE | RMSE | MAPE | MAE | RMSE | MAPE | MAE | RMSE | MAPE |
| AUTOFORMER | 0.1078 | 0.1489 | 3.4948 | 0.1936 | 0.3917 | 2.8225 | - | - | - | 0.6654 | 1.1961 | 0.3363 |
| INFORMER | 0.0827 | 0.1296 | 3.4536 | 0.1255 | 0.3183 | 2.4051 | - | - | - | 2.6893 | 4.7431 | 0.8996 |
| PYRAFORMER | 0.1451 | 0.1862 | 3.5104 | 0.0957 | 0.2651 | 2.0245 | - | - | - | 3.4571 | 5.4846 | 0.9987 |
| CROSSFORMER | 0.0858 | 0.1281 | 3.4378 | 0.1566 | 0.2927 | 2.7246 | 0.0592 | 0.0850 | 0.1335 | 2.1863 | 4.6706 | 0.5858 |
| DLINEAR | 0.0895 | 0.1351 | 3.4382 | 0.0594 | 0.3159 | 1.4073 | 0.0544 | 0.0814 | 0.1182 | 0.2045 | 0.4458 | 0.2115 |
| DCRNN | 0.4772 | 0.5995 | 3.8203 | 0.4397 | 0.5655 | 3.6791 | 0.6491 | 0.7858 | 1.1309 | 3.9790 | 5.9690 | 1.1232 |
| STGCN | 0.0873 | 0.1351 | 3.4544 | 0.0761 | 0.1901 | 1.7022 | 0.0642 | 0.0923 | 0.1472 | 3.2116 | 5.4279 | 0.8565 |
| GWNET | 0.0838 | 0.1339 | 3.4561 | 0.0513 | 0.1698 | 1.2301 | 0.0564 | 0.0833 | 0.1231 | 2.4842 | 5.0064 | 0.6153 |
| MTGNN | 0.0843 | 0.1343 | 3.4613 | 0.0518 | 0.1711 | 1.2702 | 0.0557 | 0.0833 | 0.1232 | 2.4513 | 4.2893 | 0.6835 |
| STEMGNN | 0.1558 | 0.2002 | 3.4951 | 0.2004 | 0.2977 | 3.0139 | 0.1147 | 0.1496 | 0.2577 | 3.9085 | 5.8803 | 1.1068 |
| AGCRN | 0.2169 | 0.3441 | 3.4381 | 0.5697 | 0.6508 | 3.9500 | 0.0991 | 0.1320 | 0.2286 | 3.5163 | 5.6340 | 0.9627 |
| FOURIERGNN | 0.0809 | 0.1245 | 3.4414 | 0.1040 | 0.2246 | 2.2089 | 0.0565 | 0.0879 | 0.1366 | 0.2729 | 0.5113 | 0.2345 |
| UFGTIME | 0.0809 | 0.1259 | 3.4372 | 0.0471 | 0.1696 | 0.8746 | 0.0536 | 0.0806 | 0.1173 | 0.1918 | 0.4488 | 0.2051 |

Table 3: Long-Term Multivariate Time Series Forecasting Results on Four ETT Datasets. Best Results Per Dataset Highlighted in Grey .

| DATASETS | STEPS | UFGTIME | | FOURIERGNN | | CROSSFORMER | | TIDE | | DLINEAR | | PYRAFORMER | | AUTOFORMER | | INFORMER | |
|---|---|---|---|---|---|---|---|---|---|---|---|---|---|---|---|---|---|
| | | MSE | MAE | MSE | MAE | MSE | MAE | MSE | MAE | MSE | MAE | MSE | MAE | MSE | MAE | MSE | MAE |
| ETTM1 | 96 | 0.314 | 0.358 | 0.581 | 0.462 | 0.375 | 0.415 | 0.364 | 0.387 | 0.345 | 0.372 | 0.543 | 0.510 | 0.505 | 0.475 | 0.672 | 0.571 |
| | 192 | 0.187 | 0.381 | 0.904 | 0.643 | 0.453 | 0.474 | 0.398 | 0.404 | 0.381 | 0.390 | 0.557 | 0.537 | 0.573 | 0.509 | 0.795 | 0.669 |
| | 336 | 0.369 | 0.421 | 0.919 | 0.646 | 0.548 | 0.526 | 0.428 | 0.425 | 0.414 | 0.424 | 0.754 | 0.655 | 0.621 | 0.537 | 1.212 | 0.871 |
| | 720 | 0.884 | 0.468 | 0.927 | 0.648 | 0.857 | 0.713 | 0.487 | 0.461 | 0.473 | 0.451 | 0.908 | 0.724 | 0.749 | 0.5694 | 1.307 | 0.893 |
| | Avg | 0.439 | 0.407 | 0.833 | 0.600 | 0.563 | 0.532 | 0.419 | 0.419 | 0.404 | 0.409 | 0.691 | 0.607 | 0.612 | 0.523 | 0.997 | 0.751 |
| ETTM2 | 96 | 0.081 | 0.321 | 0.574 | 0.415 | 0.267 | 0.349 | 0.207 | 0.305 | 0.195 | 0.294 | 0.435 | 0.507 | 0.255 | 0.339 | 0.365 | 0.453 |
| | 192 | 0.104 | 0.408 | 0.684 | 0.503 | 0.472 | 0.479 | 0.290 | 0.364 | 0.283 | 0.359 | 0.730 | 0.673 | 0.281 | 0.340 | 0.5334 | 0.563 |
| | 336 | 0.275 | 0.466 | 0.804 | 0.594 | 0.919 | 0.702 | 0.377 | 0.422 | 0.384 | 0.427 | 1.201 | 0.845 | 0.339 | 0.375 | 1.363 | 0.887 |
| | 720 | 0.429 | 0.575 | 0.970 | 0.705 | 4.874 | 1.601 | 0.558 | 0.524 | 0.516 | 0.502 | 3.625 | 1.451 | 0.433 | 0.432 | 3.379 | 1.338 |
| | Avg | 0.222 | 0.442 | 0.758 | 0.554 | 1.633 | 0.782 | 0.358 | 0.404 | 0.344 | 0.396 | 1.498 | 0.869 | 0.327 | 0.372 | 1.410 | 0.810 |
| ETTH1 | 96 | 0.568 | 0.421 | 0.115 | 0.495 | 0.441 | 0.457 | 0.479 | 0.464 | 0.396 | 0.430 | 0.664 | 0.612 | 0.449 | 0.459 | 0.865 | 0.713 |
| | 192 | 0.216 | 0.450 | 0.247 | 0.571 | 0.521 | 0.503 | 0.525 | 0.492 | 0.449 | 0.454 | 0.790 | 0.681 | 0.500 | 0.482 | 1.008 | 0.792 |
| | 336 | 0.739 | 0.421 | 1.173 | 0.574 | 0.659 | 0.603 | 0.569 | 0.551 | 0.487 | 0.465 | 0.891 | 0.738 | 0.521 | 0.496 | 1.107 | 0.809 |
| | 720 | 0.748 | 0.602 | 0.733 | 0.716 | 0.893 | 0.736 | 0.770 | 0.672 | 0.516 | 0.513 | 0.963 | 0.782 | 0.514 | 0.512 | 1.181 | 0.865 |
| | Avg | 0.567 | 0.473 | 0.567 | 0.589 | 0.628 | 0.574 | 0.541 | 0.507 | 0.462 | 0.466 | 0.827 | 0.703 | 0.496 | 0.487 | 1.040 | 0.794 |
| ETTH2 | 96 | 0.561 | 0.394 | 0.519 | 0.564 | 0.681 | 0.592 | 0.400 | 0.440 | 0.343 | 0.396 | 0.645 | 0.597 | 0.385 | 0.397 | 3.755 | 1.525 |
| | 192 | 0.552 | 0.455 | 0.529 | 0.638 | 1.837 | 1.054 | 0.528 | 0.509 | 0.473 | 0.474 | 0.788 | 0.683 | 0.557 | 0.511 | 5.602 | 1.931 |
| | 336 | 1.206 | 0.478 | 1.329 | 0.672 | 3.000 | 1.472 | 0.643 | 0.571 | 0.603 | 0.546 | 0.907 | 0.747 | 0.482 | 0.486 | 4.721 | 1.835 |
| | 720 | 1.225 | 0.653 | 1.257 | 0.967 | 3.024 | 1.399 | 0.874 | 0.679 | 0.812 | 0.654 | 0.963 | 0.783 | 0.515 | 0.611 | 3.647 | 1.625 |
| | Avg | 0.961 | 0.495 | 0.833 | 0.710 | 2.136 | 1.130 | 0.611 | 0.550 | 0.558 | 0.517 | 0.826 | 0.703 | 0.484 | 0.501 | 4.431 | 1.729 |

***Is UFGIME still effective in extracting long-term temporal relationships?*** It was mentioned by FourierGNN paper that graph models are more focused on dealing with dynamic patterns rather than long-range dependencies such as periodic patterns and trends. In our method, we incorporate trend

decomposition and global framelet message-passing operators to enable our models to capture global patterns, making our approach capable of performing long-term forecasting tasks. Therefore, we test our method and FourierGNN on four public long-range forecasting datasets for 96, 192, 336, and 720 steps. Comparison results of state-of-arts long-range forecasting baselines are shown in Table 3.

Surprisingly, FourierGNN does not exhibit significant performance loss with longer prediction windows, even outperforming some transformer-based methods. Compared with FourierGNN, our method shows competitive performance in long-range prediction and matches some baselines specifically designed for long-range time forecasting. This provides strong evidence that our sophisticated design for preserving global patterns benefits long-range predictions.

## 5.3 Resource Utilization Analysis

In Table 4, we compare the resource utilization of selected baselines. To eliminate the effect of hardware differences, we compare the utilization metrics generated by THOP[1], including total parameter volume and Giga-floating-point operations per second (Gflop/s). To assess scalability, we also vary the

Table 4: Comparison of Parameters and Computational Costs for Various Model Hidden Size on the **Wiki-500** Dataset with a Batch Size of 32. Computational Costs are shown in **Gflop/s**.

| BASELINES | HIDDEN 32 | | HIDDEN 64 | | HIDDEN 128 | | HIDDEN 256 | |
|---|---|---|---|---|---|---|---|---|
| | Param | Gflop/s | Param | Gflop/s | Param | Gflop/s | Param | Gflop/s |
| CROSSFORMER | $588,216$ | $30.2640$ | $1,398,616$ | $70.0281$ | $3,707,544$ | $178.0567$ | $11,077,912$ | $508.1150$ |
| STEMGNN | $1,800,504$ | $57.5376$ | $1,800,504$ | $57.5376$ | $1,800,504$ | $57.5376$ | $1,800,504$ | $57.5376$ |
| INFORMER | $183,348$ | $0.1664$ | $404,340$ | $0.3642$ | $963,060$ | $0.8542$ | $2,547,444$ | $2.2118$ |
| MTGNN | $106,268$ | $13.5128$ | $195,548$ | $26.5872$ | $374,108$ | $52.7361$ | $731,228$ | $105.0339$ |
| FOURIERGNN | $68,076$ | $2.1748$ | $70,540$ | $2.2528$ | $75,368$ | $2.4084$ | $85,324$ | $2.7197$ |
| UFGTIME | $3,690$ | $0.1245$ | $7,261$ | $0.2341$ | $14,045$ | $0.4532$ | $27,613$ | $0.8915$ |
| AGCRN | $3,960$ | $0.1232$ | $7,080$ | $0.2331$ | $13,480$ | $0.4321$ | $26,840$ | $0.8531$ |

hidden size settings. AGCRN shows outstanding efficiency, surpassing most baselines, while our method achieves comparable efficiency with only about 1/6 of the resources required by FourierGNN. In terms of scalability, aside from StemGNN, which lacks a hidden size option, all methods exhibit competitive performance, with the Transformer showing the weakest scalability.

***Is the $\mathcal{O}(NT \log(NT))$ Time Complexity Sufficiently Efficient for Pure Graph Paradigm in Time Series Forecasting?*** The FourierGNN design introduces the Discrete Fourier Transform (DFT) to reduce the complexity of the convolution operation over a fully connected graph to $\mathcal{O}(NT \log(NT)D + NTD^2)$. In our method, due to our sparse graph design, the framelet graph convolution time complexity is $\mathcal{O}(NTkD + NTD^2)$. Compared with FourierGNN, we find that the difference in complexity lies in the factors $k$ and $\log(NT)$. We conduct further simulations to determine the settings of same level of complexity. As shown in Figure 4, the surface indicates the $k$ settings to achieve the same level of complexity as FourierGNN. In practice, we set $k = 2$, which explains the outstanding efficiency of our method in Table 4.

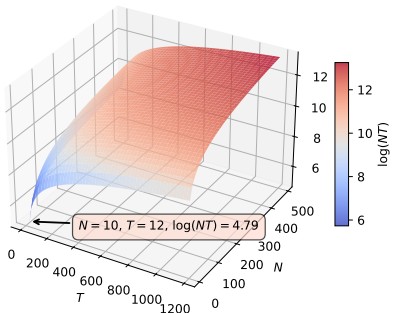

Figure 4: Complexity Boundary across Input Length $T$ and Number of Time Series $N$

## 5.4 Ablation Analysis

In this section, we conduct an ablation analysis to explore the impact of different design choices in our model. We divide the experiments into four parts to test designs such as Sparse Graph Generation(SG), Framelet Graph Convolution (FrC), Convolutions on the hyperspectral graph (GC), and Frequency Transformation to form the hyperspectral graph. We run comparisons over 10 iterations to objectively assess performance differences and perform two-way ANOVA to test the statistical significance of performance differences. The main observations are summarized below. ***Sparse or Fully Connected Graph?*** We replace the sparse graph with a fully connected graph to compare their impact. The results shown in Figure 5a indicate the superiority of sparse graphs. ***Is Framelet Necessary?*** We replace framelet convolution with GCN (Kipf & Welling, 2017), and the framelet operator shows outstanding performance, evidencing its necessity (see Figure 5b). ***Do We Really Need a Graph***

---

[1] https://github.com/ultralytics/thop

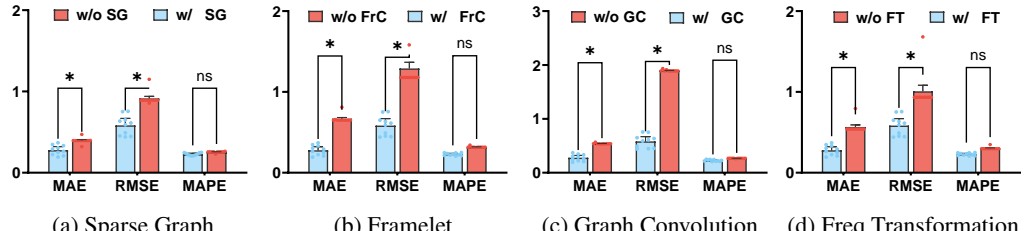

(a) Sparse Graph     (b) Framelet     (c) Graph Convolution     (d) Freq Transformation

Figure 5: Ablation Study of Key Designs of UFGTIME on **COVID-CAL** Dataset. Differences were Analyzed using a Two-Way ANOVA Test. "*" Indicate **Statistical Significant** at the 0.05 Level, While "ns" Denotes **No** Statistically Significant.

*in the Pure Graph Paradigm?* We remove the graph convolution and replace it with a linear layer to test its significance. As shown in Figure 5c, the model with graph convolution outperforms significantly, indicating the capability to capture complex temporal patterns. *hyperspectral graph vs. hypervariate Graph* Studies above suggest that the Hypervariate Graph may lose sequential information. We compare the UFGTIME model on both graphs. We aim to preserve sequential information through frequency signals by applying frequency transformations like DFT. Figure 5d supports that the hyperspectral graph is a more reasonable setting for the pure graph paradigm in time series forecasting.

## 5.5 HYPERPARAMETER SENSITIVITY ANALYSIS

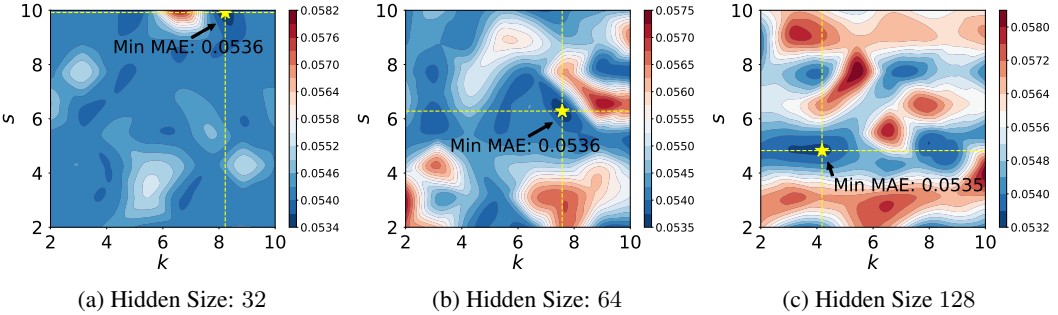

(a) Hidden Size: 32     (b) Hidden Size: 64     (c) Hidden Size 128

Figure 6: Contour Plots of Hyperparameters Surface

In this section, we conduct a sensitivity study of our proposed method. To assess the impact of the model's hyperparameters, we conduct a grid search on the **ECG** dataset over three architectural hyperparameters: framelet dilation scale $s$, number of graph neighbors $k$, and hidden size. The main observations are as follows: *Are* **UFGTIME** *Sensitive to Hyperparameters?* Based on the results in Figure 6, we find that the MAE of our model is insensitive to changes in hyperparameters, remaining around 0.055. *Patterns of Hyperparameters?* From Figure 6, we observe that our model is more sensitive to hidden size; increasing the number of hidden units sharpens the contour surface. Moreover, as the hidden size increases, the optimal values of $k$ and $s$ decrease, indicating that the model maintains an equilibrium between dilation scale, number of neighbors, and hidden size to preserve complexity.

## 6 CONCLUSION

In this work, we addressed the potential limitations of the hypervariate graph from FourierGNN by proposing a reasonable framework that embeds advanced graph operations and frequency transformations, demonstrating the feasibility of the pure graph paradigm in time series forecasting. We transformed the time series into a hyperspectral graph to preserve sequential information and replaced the fully connected graph with a sparse KNN graph for higher efficiency. To tackle short-sightedness in sparse graphs and smoothing issues in convolution on hyperspectral graph, we introduced an advanced framelet graph convolution operator that extracts both local and global temporal dependencies while alleviating smoothing. We conducted comprehensive performance and efficiency comparisons on extensive temporal datasets to evaluate our method's improvements. The results indicate our method's potential as a new solution for time series forecasting using the pure graph paradigm.

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

## A  THEORETICAL JUSTIFICATION

### A.1  PROOFS REGARDING TO THE HYPERVARIATE GRAPH LOSING SEQUENTIAL ORDER

**Propostion 1.** *The hypervariate graph is insensitive to the temporal order of the original time series, thereby discarding critical sequential information.*

*Proof.* Let $\boldsymbol{G}_1 = (\boldsymbol{X}^{\boldsymbol{G}_1}, \boldsymbol{J})$ and $\boldsymbol{G}_2 = (\boldsymbol{X}^{\boldsymbol{G}_2}, \boldsymbol{J})$ be two hypervariate graphs, where $\boldsymbol{X}^{\boldsymbol{G}_1} = \{X_1^{\boldsymbol{G}_1}, \ldots, X_n^{\boldsymbol{G}_1}\}$ and $\boldsymbol{X}^{\boldsymbol{G}_2} = \{X_1^{\boldsymbol{G}_2}, \ldots, X_n^{\boldsymbol{G}_2}\}$, with $n \in \{1, \ldots, NT\}$, representing the sets of node features that transformed from time series $\boldsymbol{X}^1, \boldsymbol{X}^2 \in \mathbb{R}^{N \times D \times T}$. We assume that the features are identical but permuted, meaning there exists a permutation $\rho : \{1, 2, \ldots, n\} \mapsto \{1, 2, \ldots, n\}$ such that for all $i \in \{1, 2, \ldots, n\}$, we have $X_i^{\boldsymbol{G}_1} = X_{\rho(i)}^{\boldsymbol{G}_2}$.

By definition of graph isomorphism, if there exists such a permutation $\rho$ mapping the node features of $\boldsymbol{G}_1$ to those of $\boldsymbol{G}_2$, then the graphs $\boldsymbol{G}_1$ and $\boldsymbol{G}_2$ are isomorphic, i.e., $\boldsymbol{G}_1 \cong \boldsymbol{G}_2$.

However, the original time series $\boldsymbol{X}^1$ and $\boldsymbol{X}^2$ are different ($\boldsymbol{X}^1 \neq \boldsymbol{X}^2$), yet their corresponding hypervariate graphs $\boldsymbol{G}_1$ and $\boldsymbol{G}_2$ are isomorphic. This implies that the hypervariate graph representation does not retain the sequential order information of the underlying time series. Specifically, different time series can lead to isomorphic hypervariate graphs if the node order is permuted, thus discarding crucial temporal information.

$\square$

## B  ADDITIONAL DETAILS

### B.1  POLYNOMIAL APPROXIMATION OF FRAMELET DECOMPOSITION

To avoid time-consuming eigendecomposition, the Chebyshev polynomial approximation provides an efficient and scalable solution for framelet decomposition (Dong, 2017; Zheng et al., 2021). Let $K$ denote the highest order of the involved Chebyshev polynomial. The $K$-order approximations of $\alpha$ and $\{\beta^{(r)}\}_{r=1}^L$ are denoted as $\mathcal{T}_0^K$ and $\{\mathcal{T}_r^K\}_{r=1}^L$, respectively. The approximated framelet decomposition operator $\mathcal{W}_{r,j}$ is defined as the product of Chebyshev polynomials of the graph Laplacian $\mathcal{L}$, i.e.,

$$\mathcal{W}_{r,j} = \begin{cases} \mathcal{T}_0^K(2^{-m}\mathcal{L}) & j = 1, \\ \mathcal{T}_r^K(2^{-(m+j-1)}\mathcal{L})\mathcal{T}_0^K(2^{-1(m+j-2)}\mathcal{L}) \ldots \mathcal{T}_0^K(2^{-m}\mathcal{L}) & j = 2, \ldots, J \end{cases} \quad (10)$$

Here we denote the operation $2^{-m}\mathcal{L}$ means a scaling operation onto Laplacian's eigenvalues. The real-valued dilation scale $m$ is the smallest integer such that $\lambda_{max} = \lambda_N \leq \pi$, so that the range of $\pi$ fits the domain of Chebyshev polynomial.

### B.2  ADDITIONAL DETAILS OF FFN

The fully connected feed-forward network (FFN) consists of two linear transformations with SiLU activation and InstantNorm2D in between. Suppose the hidden unit size is set to feature size $F$. The FFN is formulated as follows:

$$\boldsymbol{H}_t = \text{Reshape}(\boldsymbol{H}_t)[N, D, T]$$

$$\boldsymbol{H}_t^1 = \text{SiLU}\left(\text{InstantNorm2D}(\boldsymbol{H}_t)\right)\boldsymbol{W}_1 \quad (11)$$

$$\boldsymbol{H}_t^2 = \text{SiLU}\left(\text{InstantNorm2D}(\boldsymbol{H}_t^1)\right)\boldsymbol{W}_2,$$

where $\boldsymbol{W}_1 \in \mathbb{R}^{T \times F}$ and $\boldsymbol{W}_2 \in \mathbb{R}^{F \times F}$ are weight matrices. After getting reshaped trend information $\boldsymbol{T}_t \in \mathbb{R}^{N \times D \times T}$, we conduct a trend information fusion with a single linear layer as:

$$\widehat{Y}_{t+1} = \left(\boldsymbol{H}_t^2 + \boldsymbol{T}_t\boldsymbol{W}_{trend}\right)\boldsymbol{W}_3, \quad (12)$$

where $\boldsymbol{W}_{trend} \in \mathbb{R}^{T \times F}$ is weight matrix that embed trend information into hidden space, and $\boldsymbol{W}_3 \in \mathbb{R}^{F \times \tau}$ is weight matrix that project hidden state into prediction length $\tau$.

### B.3 DETAILS OF COMPLEXITY ANALYSIS

For simplicity of analysis, we assume the weights of our Fourier domain framelet message-passing operator are in $\mathbb{C}^{D \times D}$, and the frequency signal has the same length $T$ as the time series. The time complexity of a single layer of our framelet operator on the hyperspectral graph $G^S$ is given by $\mathcal{O}(L(J+1)(NT)^2 D + NTD^2)$, where $L$ represents the number of high-pass filters and $J$ stands for the levels of the scaling function. Since the hyperspectral graph is sparse, the complexity term of graph convolution $NT$ can be replaced by $|\mathcal{E}|$, the total number of edges in the hyperspectral graph. Additionally, the topology of the hyperspectral graph is generated using the KNN algorithm, resulting in $|\mathcal{E}| \leq kNT$, where $k$ is the number of neighbours. Therefore, an upper bound for the complexity of our framelet operator is $\mathcal{O}(L(J+1)NTkD + NTD^2)$. In practice, we set both $L$ and $J$ to 1, making the product $L(J+1)$ a small constant that can be omitted from the total complexity. Finally, the complexity of the framelet operator simplifies to $\mathcal{O}(NTkD + NTD^2)$, resulting in UFGTIME being highly efficient.

### B.4 DETAILS OF DATASETS

Table 5: Summary of datasets

| Datasets | SOLAR-FL | WIKI-500 | TRAFFIC | ECG | ELECTRICITY2H | COVID-CAL | ETTM1/2 | ETTH1/2 |
|---|---|---|---|---|---|---|---|---|
| Samples | 4380 | 803 | 10560 | 4999 | 4380 | 345 | 69680 | 17420 |
| Variables | 593 | 500 | 963 | 140 | 370 | 60 | 7 | 7 |
| Granularity | 2 hour | 1 day | 1 hour | - | 2 hour | 1 day | 15 minutes | 1 hour |
| Start time | 2006-01-01 | 2015-01-07 | 2015-01-01 | - | 2014-01-01 | 2020-01-22 | 2016-07-01 | 2016-07-01 |

We follow the instructions provided by FourierGNN to set up six short-term forecasting datasets from the source (refer to dataset detail in Table 5). However, we found that some datasets downloaded from these sources did not match the data descriptions in the FourierGNN paper. Therefore, we provide the full details of the data sources and processing steps for the datasets used in our paper, as follows:

- ECG[2]: This dataset originally from a 20-hour heartbeat recording and 5,000 heartbeat random sampling are made during data generation, the ECG dataset lacks granularity and start time information.

- SOLAR-FL[3]: We followed the FourierGNN setup to select the Florida subset of the solar dataset from the eastern states. However, we found that the dataset was originally recorded at 15-minute intervals and not match the 3,650 samples reported in FourierGNN. Therefore, we down-sampled it to 2-hour intervals, resulting in a total of 4,380 samples.

- COVID-CAL[4]: The source repository contains multiple COVID-19 datasets, and there is no specific information on which dataset FourierGNN used. We selected the CSSE COVID-19 confirmed cases dataset for the U.S. to best approximate the original dataset used in FourierGNN. We filtered hospital records from 60 counties in California, resulting in a total of 345 timestamps.

- ELECTRICITY2H[5]: We downloaded the Electricity dataset, which contains 140,211 samples recorded at 15-minute intervals. We down sampled the data to 2-hour intervals to create the ELECTRICITY2H dataset.

- WIKI-500[6]: The original Wiki dataset contains 145,000 samples across 830 timestamps. We randomly selected 500 time series to form the WIKI-500 dataset.

- TRAFFIC[7]: This dataset contains hourly traffic data from 963 freeway sensors in San Francisco. The traffic data are collected starting from 2015/01/01 at 1-hour intervals.

---

[2] https://timeseriesclassification.com/description.php?Dataset=ECG5000
[3] https://www.nrel.gov/grid/solar-power-data.html
[4] https://github.com/CSSEGISandData/COVID-19
[5] https://archive.ics.uci.edu/dataset/321/electricityloaddiagrams20112014
[6] https://drive.google.com/uc?export=download&id=1VytXoL_vkrLqXxCR5IOXgE45hN2UL5oB
[7] https://drive.google.com/uc?export=download&id=1dyeYj8IJwZ3bKvk1H67eaDTANdapKe7w

For long-range time series forecasting dataset ETTH1/2 and ETTM1/2, we follow the original setting provided by Zhou et al. (2021) and details are shown as Table 3

## B.5 ADDITIONAL EXPERIMENT RESULTS

Due to space limitations, the complete short-term forecasting results are presented below:

Table 6: Full Short-Term Forecasting Results on Six Datasets. Best and Second Best Results Per Dataset Highlighted in Grey and Underlined Respectively. ECG Results for Partial Transformer-based Methods are Denoted as '−' Due to Missing Temporal Information.

| BASELINES | SOLAR-FL | | | WIKI-500 | | | TRAFFIC | | |
|---|---|---|---|---|---|---|---|---|---|
| | MAE | RMSE | MAPE | MAE | RMSE | MAPE | MAE | RMSE | MAPE |
| AUTOFORMER | 0.1078 | 0.1489 | 3.4948 | 0.1936 | 0.3917 | 2.8225 | 0.0669 | 0.1018 | 0.9734 |
| INFORMER | 0.0827 | 0.1296 | 3.4536 | 0.1255 | 0.3183 | 2.4051 | 0.0522 | 0.0837 | 0.6640 |
| PYRAFORMER | 0.1451 | 0.1862 | 3.5104 | 0.0957 | 0.2651 | 2.0245 | 0.0466 | 0.0768 | 0.6961 |
| CROSSFORMER | 0.0858 | 0.1281 | 3.4378 | 0.1566 | 0.2927 | 2.7246 | 0.0642 | 0.0940 | 1.0889 |
| DLINEAR | 0.0895 | 0.1351 | 3.4382 | 0.0594 | 0.3159 | 1.4073 | 0.0655 | 0.1036 | 0.9161 |
| DCRNN | 0.4772 | 0.5995 | 3.8203 | 0.4397 | 0.5655 | 3.6791 | 0.4404 | 0.5507 | 3.2122 |
| STGCN | 0.0873 | 0.1351 | 3.4544 | 0.0761 | 0.1901 | 1.7022 | 0.0356 | 0.0619 | 0.5107 |
| GWNET | 0.0838 | 0.1339 | 3.4561 | 0.0513 | 0.1698 | 1.2301 | 0.0354 | 0.0638 | 0.5164 |
| MTGNN | 0.0843 | 0.1343 | 3.4613 | 0.0518 | 0.1711 | 1.2702 | 0.0348 | 0.0618 | 0.4972 |
| STEMGNN | 0.1558 | 0.2002 | 3.4951 | 0.2004 | 0.2977 | 3.0139 | 0.0694 | 0.1028 | 1.0486 |
| AGCRN | 0.2169 | 0.3441 | 3.4381 | 0.5697 | 0.6508 | 3.9500 | 0.0973 | 0.1336 | 1.4485 |
| FOURIERGNN | 0.0809 | 0.1245 | 3.4414 | 0.1040 | 0.2246 | 2.2089 | 0.0403 | 0.0696 | 0.5908 |
| UFGTIME | 0.0809 | 0.1259 | 3.4372 | 0.0471 | 0.1696 | 0.8746 | 0.0351 | 0.0618 | 0.5191 |

| BASELINES | ECG | | | ELECTRICITY2H | | | COVID-CAL | | |
|---|---|---|---|---|---|---|---|---|---|
| | MAE | RMSE | MAPE | MAE | RMSE | MAPE | MAE | RMSE | MAPE |
| AUTOFORMER | - | - | - | 0.0961 | 0.1273 | 0.4902 | 0.6654 | 1.1961 | 0.3363 |
| INFORMER | - | - | - | 0.1241 | 0.1611 | 0.6116 | 2.6893 | 4.7431 | 0.8996 |
| PYRAFORMER | - | - | - | 0.1525 | 0.1986 | 0.8710 | 3.4571 | 5.4846 | 0.9987 |
| CROSSFORMER | 0.0592 | 0.0850 | 0.1335 | 0.1403 | 0.1750 | 0.8241 | 2.1863 | 4.6706 | 0.5858 |
| DLINEAR | 0.0544 | 0.0814 | 0.1182 | 0.0859 | 0.1193 | 0.4912 | 0.2045 | 0.4458 | 0.2115 |
| DCRNN | 0.6491 | 0.7858 | 1.1309 | 0.5532 | 0.6879 | 1.8591 | 3.9790 | 5.9690 | 1.1232 |
| STGCN | 0.0642 | 0.0923 | 0.1472 | 0.1155 | 0.1587 | 0.6625 | 3.2116 | 5.4279 | 0.8565 |
| GWNET | 0.0564 | 0.0833 | 0.1231 | 0.0782 | 0.1201 | 0.4536 | 2.4842 | 5.0064 | 0.6153 |
| MTGNN | 0.0557 | 0.0824 | 0.1222 | 0.0834 | 0.1235 | 0.5106 | 2.4513 | 4.2893 | 0.6835 |
| STEMGNN | 0.1147 | 0.1496 | 0.2577 | 0.2929 | 0.3598 | 1.0889 | 3.9085 | 5.8803 | 1.1068 |
| AGCRN | 0.0991 | 0.1320 | 0.2286 | 0.1735 | 0.2193 | 0.9563 | 3.5163 | 5.6340 | 0.9627 |
| FOURIERGNN | 0.0565 | 0.0879 | 0.1366 | 0.0927 | 0.1359 | 0.5589 | 0.2729 | 0.5113 | 0.2345 |
| UFGTIME | 0.0536 | 0.0806 | 0.1173 | 0.0752 | 0.1164 | 0.0455 | 0.1918 | 0.4488 | 0.2051 |

## B.6 REPRODUCTION DETAILS

In what follows, we present the search space of hyperparameters for our proposed UFGTIME and compared baselines. To ensure that all models are evaluated under a fair comparison environment, we perform hyperparameter tuning of all models under the same experimental framework and designate

the same search space of common hyperparameters (i.e. Learning rate, Weight decay and Batch size). We denote {} as a set with discrete values and [] as a closed interval containing continuous values.

**UFGTIME**

- Learning rate $\in$ [1e-5, 5e-2]
- Hidden size $\in$ {16, 32, 64, 128, 256}
- Weight decay: wd $\in$ [1e-5, 5e-2]
- Chebyshev order $\in$ {2, 3, 4}
- K neighbors $\in$ {1-10} with a step of 1
- s $\in$ [1.1, 10.0]
- Batch size $\in$ {16, 32, 64, 128, 256}
- epochs: 100

### B.6.1 BASELINES

In experiments, we consider 12 baselines for the validation of our proposed UFGTIME and provide their brief introduction as well as corresponding hyperparameters considered in our implementations as follows.

**Autoformer** (Wu et al., 2021) Comprising a decomposition architecture with an Auto-correlation mechanism, Autoformer is able to handle complex time series with progressive decomposition capacities and capture dependencies at the sub-series level. We obtain the source code from `https://github.com/thuml/Autoformer`

- d_model $\in$ {256, 512}
- d_ff $\in$ {512, 1024, 2048}
- n_head $\in$ {6, 8, 10}

**Informer** (Zhou et al., 2021) Aiming to improve prediction capacity for long sequence time-series forecasting, Informer is constructed with three distinctive modules that achieve lower time complexity, effectively handle extremely long input sequences, and improve the inference speed of long-sequence predictions respectively. We obtain the source code from `https://github.com/zhouhaoyi/Informer2020`

- d_model $\in$ {256, 512}
- d_ff $\in$ {512, 1024, 2048}
- n_head $\in$ {6, 8, 10}

**Pyraformer** (Liu et al., 2021) Pyraformer considers the multi-resolution representation of the time series using the pyramidal attention module. This module introduces an inter-scale tree structure to capture features at different resolutions and also an intra-scale neighboring connection to capture the temporal dependencies. We obtain the source code from `https://github.com/ant-research/Pyraformer`

- d_model $\in$ {256, 512}
- d_inner_hid $\in$ {256, 512}
- n_head $\in$ {4, 6}

**Crossformer** (Zhang & Yan, 2023) Going beyond modeling the temporal dependency, Crossformer further considers the dependency among different variables for multivariate time series forecasting. It utilizes the Dimension-Segment-Wise embedding and a Two-Stage Attention layer to model both dependencies across time and dimension. We obtain the source code from `https://github.com/Thinklab-SJTU/Crossformer`

- d_model $\in$ {256, 512}
- d_inner_hid $\in$ {256, 512}

- n_head $\in \{6, 8, 10\}$

**DLinear** (Zeng et al., 2023) DLinear is a simple one-layer linear model that regresses historical time series to conduct forecasts directly. The design of this model aims to retrieve the loss information from the nature of the permutation invariant self-attention mechanism of Transformers, which preserves order information through positional encoding. We obtain the source code from `https://github.com/honeywell21/DLinear`

**TiDE** (Das et al., 2023) TiDE is an MLP-based encoder-decoder model that shows high simplicity and is able to capture covariates and non-linear dependencies. It encodes historical data and decodes data with future covariates using dense MLPs. We obtain the source code from `https://github.com/google-research/google-research/tree/master/tide`

- d_model $\in \{256, 512\}$
- d_inner_hid $\in \{256, 512\}$
- e_layers $\in \{2\}$
- d_layers $\in \{2\}$

**DCRNN** (Li et al., 2018) Integrating recurrent neural networks, DCRNN considers a bidirectional graph random work technique to capture spatial relationships for modeling temporal dynamics. DCRNN demands a pre-defined graph adjacency matrix and we utilize K-nearest neighbors with $k = 10$ to generate corresponding graph structures. We obtain the source code from the BasicTS+(Shao et al., 2023a): `https://github.com/GestaltCogTeam/BasicTS/tree/master/baselines/DCRNN/arch`

- Number of rnn layers $\in \{2, 3, 4\}$
- Rnn units $\in \{32, 64, 128, 256\}$
- Use curriculum learning $\in \{$True, False$\}$

**STGCN** (Yu et al., 2018) By formulating the problem on graphs, STGCN simultaneously captures spatial and temporal correlations through the integration of graph convolution and gated temporal convolution. STGCN demands a pre-defined graph adjacency matrix and we utilize K-nearest neighbors with $k = 10$ to generate corresponding graph structures. We obtain the source code from the BasicTS+(Shao et al., 2023a): `https://github.com/GestaltCogTeam/BasicTS/tree/master/baselines/STGCN/arch`

- Kt $\in \{3\}$
- Ks $\in \{3\}$
- blocks $\in \{[[1], [64, 16, 64], [64, 16, 64], [128, 128], [12]]\}$
- activation function $\in \{$glu, gtu$\}$

**GWNET** (Wu et al., 2019) GWNET learns an adaptive dependency matrix through node embedding to capture graph hidden spatial dependency. We obtain the source code from the BasicTS+(Shao et al., 2023a): `https://github.com/GestaltCogTeam/BasicTS/tree/master/baselines/GWNet/arch`

- Residual channels $\in \{32, 64\}$
- Dilation channels $\in \{32, 64\}$
- Skip channels $\in \{128, 256\}$
- End channels $\in \{256, 512\}$
- Kernel size $\in \{2, 3, 4\}$
- Blocks $\in \{3, 4, 5\}$
- Layers $\in \{2, 3, 4\}$

**MTGNN** (Wu et al., 2020) MTGNN automatically extracts the inherent dependency relationship and utilizes a mix-hop propagation layer along with a dilated inception layer to model spatial and temporal correlations. We obtain the source code from the BasicTS+(Shao et al., 2023a): `https://github.com/GestaltCogTeam/BasicTS/tree/master/baselines/MTGNN/arch`

- Subgraph size $\in \{10, 20, 30\}$
- Convolution channels $\in \{16, 32, 64\}$
- Residual channels $\in \{32, 64\}$
- Skip channels $\in \{64, 128\}$
- End channels $\in \{128, 256\}$
- Layers $\in \{2, 3, 4\}$

**StemGNN** (Cao et al., 2020) StemGNN leverages Graph Fourier Transform and Discrete Fourier Transform so that it is able to model inter-series correlations and temporal dependencies jointly in the spectral domain. We obtain the source code from the BasicTS+(Shao et al., 2023a): `https://github.com/GestaltCogTeam/BasicTS/tree/master/baselines/StemGNN/arch`

- Stack count $\in \{2, 3, 4\}$
- Multi-layer $\in \{3, 4, 5, 6, 7\}$

**AGCRN** (Bai et al., 2020) AGCRN captures node-specific patterns and the inter-dependencies respectively by a node parameter learning module and graph generation module. These two modules are designed in an adaptive manner that can automatically capture fine-grained spatial and temporal correlations between time-series. We obtain the source code from the BasicTS+(Shao et al., 2023a): `https://github.com/GestaltCogTeam/BasicTS/tree/master/baselines/AGCRN/arch`

- Rnn units $\in \{32, 64, 128, 256\}$
- Layer number $\in \{2, 4, 6\}$
- Chebyshev order $\in \{2, 3, 4\}$

**FourierGNN** (Yi et al., 2024) FourierGNN rethinks multivariate time series into a pure graph problem where each series value can be regarded as a graph node and performs message-passing in Fourier space such that an adequate expressiveness and lower complexity can be achieved. We obtain the source code from: `https://github.com/aikunyi/FourierGNN`

- Hidden size $\in \{128, 256, 512\}$
- Embedding size $\in \{128, 256, 512\}$

