# OpenReview forum: "UFGTime: Reforming the Pure Graph Paradigm for Multivariate Time Series Forecasting in the Frequency Domain"
_ICLR.cc/2025/Conference — ICLR 2025 Conference Withdrawn Submission_

### Official Review · Reviewer_5335 · 2024-10-17

**Soundness:** 1
**Presentation:** 2
**Contribution:** 1
**Rating:** 1
**Confidence:** 4

**Summary:**

The paper introduces a graph representation for groups of multivariate time series and a time series forecasting model based on this representation. The main motivation behind the method is to address limitations in a specific previous work (FourierGNN), which represents the input time series as a fully connected graph where each node is an observation in time and space. This representation clearly has the drawback of discarding the temporal ordering of the observations. The introduced method attempts this issue, but such a limitation is specific to FourierGNN and does not apply to a broader body of literature operating in similar settings. There are also some issues with the soundness of the method and of the empirical evaluation.

**Strengths:**

* Developing effective deep learning architectures for time series forecasting is an important research direction, and graph-based approaches are appealing in many settings.

**Weaknesses:**

There are major weaknesses that prevent me from recommending acceptance.

* **Soundness of the approach.** The proposed approach builds a graph representation by considering the K-nearest neighbors of a representation of the input obtained through an FFT along the temporal dimension. However, in such a representation, each node corresponds to a different frequency and entity. I don't understand why one would want to connect nodes that correspond to different frequencies. For example, with such a representation, signals that have all of their spectrum concentrated at totally different frequencies would get connected. This representation does not preserve the structure of the data, similarly to the method the paper is trying to fix.
* **Poor novelty and weak motivation.** As noted, the limitation the paper aims to address is specific to a single model. Addressing such a narrow issue isn't sufficient when most state-of-the-art approaches do not have this drawback. Many spatiotemporal graph neural networks utilize sound graph representations of the input, and in many cases, the processing of spatial and temporal dimensions is well integrated (e.g., see [1]). Additionally, several signal processing methods operate on pure graph representations of sets of time series. For instance, [2] uses product graph representations. The visibility graph is another principled graph representation of input time series (e.g., see [3]). All of these representations avoid the issues the proposed method attempts to solve and are more sound than the one presented in the paper. Lastly, the novelty of the graph framelet operator is limited, as it appears to be a straightforward adaptation of an existing method.
* **Soundness of the empirical evaluation.** The empirical evaluation has some issues. Firstly, there is a growing consensus that the ETT benchmarks used for the experiments in Tab 3 are not significant for evaluating deep learning methods for time series forecasting. For example, in [4], a simple autoregressive linear model trained using ordinary least squares achieves better results than those reported in the table in almost all the considered scenarios. Secondly, some of the results in Table 2 involve baselines that require a predefined input graph, which, according to the appendix, is a kNN graph obtained in an unclear way (e.g., which similarity metric was used to create this graph?).

Given these issues, the paper is far below the acceptance threshold by ICLR standards.

Minor comments

* There must be some reporting errors in Table 3. Looking at the MSE for UFG in ETTH1, the MSE at 192 steps is lower than the MSE at 96, which does not make any sense if the testing is done properly. Differently, the MAE increases as one would expect.

[1] Wu et al., "Traversenet: Unifying space and time in message passing for traffic forecasting" TNNLS 2022\
[2] Sabbaqi et al., "Graph-time convolutional neural networks: Architecture and theoretical analysis." PAMI 2023.\
[3] Lacasa et al., "From time series to complex networks: The visibility graph" PNAS 2008\
[4] Toner et al., "An Analysis of Linear Time Series Forecasting Models." ICML 2024\

**Questions:**

--

---

### Official Review · Reviewer_iKzu · 2024-10-30

**Soundness:** 2
**Presentation:** 3
**Contribution:** 2
**Rating:** 5
**Confidence:** 4

**Summary:**

This paper introduces the hyperspectral graph structure to enhance the ability of GNNs in capturing temporal dependencies for multivariate time series forecasting. The authors proposes UFGTIME, which transforms time series data into a frequency domain representation, preserves sequential information, and constructs a sparse graph with signal similarity using KNN. This graph structure is combined with a framelet message passing mechanism, aiming to improve the ability to capture both global and local temporal patterns without over-smoothing.

**Strengths:**

+ The work presents a feasible way of addressing the limitations of hypervariate graphs by transforming time series into the frequency domain and constructing hyperspectral graphs. The use of frequency-based graph representation is an interesting twist on how spatio-temporal dependencies are handled.
+ This paper introduces a well-defined framework with innovative techniques, such as the graph framelet message passing operator. The theoretical analysis and complexity evaluations provide a thorough understanding of the approach.
+ The technical descriptions of the novel hyperspectral graph and framelet message passing are articulated well.
+ The work addresses practical limitations in a recent GNN-based time series forecasting method, proposing improvements that could lead to better generalization in various time series forecasting domains, including both short- and long-term applications.

**Weaknesses:**

+ The core motivation of this work feels narrow. While improving upon FourierGNN is a valuable contribution, the paper does not clearly establish why the disjoint modeling architecture in existing GNN-based methods fails critically in practice, or why FourierGNN's approach is inherently superior. A stronger emphasis on the overarching challenges of multivariate time series forecasting would better justify the significance of this work.
+ The storytelling could be improved, particularly in the introduction. The two key challenges in FourierGNN are not clearly articulated, and after reading the introduction, it is unclear if this research addresses a significant problem.
+ The empirical validation in Sec. 3 is somewhat weak in certain aspects. For instance, the experimental setup for the evaluation in Tab. 1 is unclear, and the performance degradation after permutation looks significant without compared to benchmarking methods, leaving doubt about the strength of the claims. Additionally, the visualization results in Fig. 2 show minimal differences between sparse and fully connected Laplacian patterns without the attention weights.
+ The construction of the hyperspectral graph closely mirrors that of the hypervariate graph, with the key distinction being that it bypasses direct modeling of sequential information by transforming time series into discrete frequency components.
+ There is a lack of comparative discussion against competitive time series models, particularly recent approaches like PatchTST and TimeMixer. The advantages of UFGTIME over these models are not sufficiently explored, which reduces the clarity around the unique contributions of this research.

**Questions:**

+ For the results in Sec. 3.2, what temporal characteristics were used in constructing the Laplacian matrices? Additionally, what are the attention weights applied in this context?
+ I am curious about the necessity of this method—what are its key advantages compared to competitive time series models like PatchTST, TimeMixer, and a recent study https://arxiv.org/pdf/2403.14587?

---

### Official Review · Reviewer_WRAi · 2024-11-01

**Soundness:** 3
**Presentation:** 2
**Contribution:** 2
**Rating:** 5
**Confidence:** 4

**Summary:**

The authors present a **GNN-based model tailored to time series forecasting** without an essentially existent aprior graph structure. They focus on overcoming the limitations of a recent state-of-the-art method that considers hypervariate (fully connected) graphs by highlighting the importance of capturing the most *crucial inter- and intra-series correlations* by incorporating some sparsity. They thus propose a method that extracts a KNN-based graph structure built upon representation extracted by the Fourier transform. Their introduced (hyperspectral) graph structure is then processed by the so-called *global Fourier framelet message-passing operator* to capture global patterns from the time series examples. After the **spectral graph embedding module**, the inverse Fourier transform returns the representation in the original time dimension, followed by a two-layer feed-forward network to predict the output vector (on the future steps) combined with the *trend embedding* of the original signal. The authors evaluate the performance of their proposed method against popular graph-based and non-graph-based models in forecasting and presenting properties of the critical components of their model through ablation studies.

**Strengths:**

Strong points of the presented work are the following:
- **[S1]:** The authors showcase the importance of spectral graphs for time series forecasting while highlighting the importance of sparsity in the structure.
- **[S2]:** The proposed framework remains simple in its design.
- **[S3]:** Experimental results show the very competitive performance of the proposed method in terms of MSE with popular baselines.
- **[S4]:** Ablation studies highlight the importance of performance when learning a (sparse) graph structure on underlying dependencies in forecasting.

**Weaknesses:**

- **[W1]:** Performance improvements achieved by the proposed method are mostly minor compared with FourierGNN for the datasets considered in short-term forecasting.
- **[W2]:** The proposed method is outperformed by baselines on datasets used for long-term forecasting. Interestingly, in several cases, there is a discrepancy in the performances achieved in terms of MSE and MAE (e.g., MAE scores of the proposed UFGTIME on ETTh2 are the best against baselines, but corresponding MSEs are almost double in value compared to the Autoformer).
- **[W4]:** The authors focus on the challenges of the pure graph paradigm, yet most graph-based methods for time series before this work [1] were similarly using KNN [3] or differentiable methods (Gumbel softmax) [2] to achieve graph sparsity. Additionally, embeddings were learned based on time series inter-variable correlations or temporal dynamics [2] or as node embeddings [3]. The paper misses an extensive discussion, e.g., in the related work section, on the critical characteristics of the graph modules introduced in the literature to adequately position the proposed UFGTIME's contribution.
- **[W5]:** The learned graph dependencies are not evaluated qualitatively. Similar to the studies/visualizations for the graph learning module in [3], it would be interesting to assess the validity of learned time series dependencies produced by the proposed graph module.
- **[W6]:** The proposed method is quite efficient regarding memory/time cost. However, it would be interesting for the discussion to be enhanced with explanations of the complexities of all considered GNN-based baselines. It is unclear to which extent the memory complexity/efficiency of the proposed method is affected by the window size and number of variables in the input, which could provide additional ablation studies.

[1] Yi, K., Zhang, Q., Fan, W., He, H., Hu, L., Wang, P., ... & Niu, Z. (2024). FourierGNN: Rethinking multivariate time series forecasting from a pure graph perspective. Advances in Neural Information Processing Systems, 36.

[2] Shang, C., Chen, J., & Bi, J. (2021). Discrete graph structure learning for forecasting multiple time series. arXiv preprint arXiv:2101.06861.

[3] Wu, Z., Pan, S., Long, G., Jiang, J., Chang, X., & Zhang, C. (2020, August). Connecting the dots: Multivariate time series forecasting with graph neural networks. In Proceedings of the 26th ACM SIGKDD international conference on knowledge discovery & data mining (pp. 753-763).

**Questions:**

Based on the above weaknesses, the following aspects of this work could benefit from additional explanations/extensions:
- **[Q1] Discrepancies in MSE/MAE performances for different datasets/horizons:** Based on **[W2]**, could you explain why the model for specific datasets performs significantly worse in terms of MSE in comparison to baselines but has the best MAE values.
- **[Q2] Visualizations for predictions/learned graph dependencies:** To address discrepancies in scores, some visualizations of predictions (true and predicted time series) corresponding to such cases (low MAE, large MSE), but also, in general, are a crucial qualitative way to access performance variations between models/datasets. Based on **[W5]**, could you provide some visualizations on the learned dependencies provided by the graph module of UFGTIME for some real-world datasets?
- **[Q3] Proper comparison of different graph learning modules:** The paper could improve its discussion of related works by differentiating between pure graph paradigms and methods that produce sparse graphs in the context of GNN architectures.
- **[Q4] Complexity analysis of GNN-based graph learning baselines:** The paper and results could benefit from a discussion on the complexity of different GNN-based baselines along different graph learning modules beyond the already mentioned FourierGNN.
- **[Q5] Impact of window size and input variables on memory complexity:** including analysis/ablations that address these factors could provide a better understanding of the scalability and resource requirements of UFGTIME for real-world time series datasets.
- **[Q6] Misc:** No standard deviations/averaging across multiple runs for the provided performance results are mentioned. Is it possible to demonstrate that the small enhancements observed across multiple cases are statistically significant compared to the best competitor?

---

### Official Review · Reviewer_HyX9 · 2024-11-04

**Soundness:** 1
**Presentation:** 1
**Contribution:** 1
**Rating:** 3
**Confidence:** 4

**Summary:**

The paper presents a time-series forecasting model that first computes the Fourier transform for individual components of the time series. It then constructs a KNN graph using triples as nodes -- each of which consisting of a time-series index, feature dimension, and frequency. Finally, the model employs framelets to make predictions.

**Strengths:**

- The paper addresses limitations of a previously published method, FourierGNN.

**Weaknesses:**

- The presentation is confusing, often blending obvious claims with unsupported statements and using ambiguous terminology.
- The paper tackles challenges that seem obvious and already addressed in the literature: (1) the importance of maintaining temporal order in time-series processing, and (2) the unsuitability of fully connected graphs in graph-based processing.
- As far as I could understand, the paper’s original contribution is limited to constructing a KNN graph from a spectral representation of the original multivariate time series.

**Questions:**

- Could you explain why lines 43-45 present a contradiction? Some papers do consider product graphs (eg https://arxiv.org/abs/2206.15174). Moreover, addressing temporal and spatial (graph) processing in subsequent steps does not seem to reduce expressiveness (see eg https://arxiv.org/abs/2103.07016).
- Lines 92-93. Claiming that CNN cannot model spatial information is simply wrong; see e.g. the field of Geospatial Deep Learning.
- Definition 1. Why is this structure called a graph if the only information is in the node features? Also, why is it called "hypervariate" if it has $D$ features, as per the original time series?
- Sections 3.1 and 3.2. These sections describe well-known facts: a fully-connected graph is regular and each permutation leads to an automorphism.  Is there an additional insight in these sections that I may have missed?
- Definition 2. Similarly to Definition 1, why is this called "hyperspectral"? What makes it “hyper”?
- Section 4.1.  As a reference, could you indicate a few state-of-the-art approaches within the "pure graph paradigm"? Currently, I only see only FourierGNN being mentioned.
- Line 212. The elements can only be rearranged as long as they are labeled or indexed by the corresponding frequency-node pair. This means they can be stored in memory in any order, but their indexing cannot be ignored. Therefore, the statement in line 215 does not logically follow from line 212. Is there an argument to support the conclusion in lines 215-217?
- What is novel in the proposed use of framelets compared to existing literature?
- Could you report standard deviations alongside the results presented?
- Figure 6. What does a contour plot represent for discrete variables? Additionally, how can the optimal $k$ be a non-integer value?

---

### Note · Authors · 2024-11-15

I have read and agree with the venue's withdrawal policy on behalf of myself and my co-authors.